# APP in the Neuromuscular Junction for the Development of Sarcopenia and Alzheimer’s Disease

**DOI:** 10.3390/ijms24097809

**Published:** 2023-04-25

**Authors:** Min-Yi Wu, Wen-Jun Zou, Daehoon Lee, Lin Mei, Wen-Cheng Xiong

**Affiliations:** 1Department of Neurosciences, School of Medicine, Case Western Reserve University, Cleveland, OH 44106, USA; 2Louis Stokes Cleveland Veterans Affairs Medical Center, Northeast Ohio VA Healthcare System, Cleveland, OH 44106, USA

**Keywords:** Alzheimer’s disease, sarcopenia, amyloid precursor protein, neuromuscular junction, neurodegenerative diseases

## Abstract

Sarcopenia, an illness condition usually characterized by a loss of skeletal muscle mass and muscle strength or function, is often associated with neurodegenerative diseases, such as Alzheimer’s disease (AD), a common type of dementia, leading to memory loss and other cognitive impairment. However, the underlying mechanisms for their associations and relationships are less well understood. The *App*, a Mendelian gene for early-onset AD, encodes amyloid precursor protein (APP), a transmembrane protein enriched at both the neuromuscular junction (NMJ) and synapses in the central nervous system (CNS). Here, in this review, we highlight APP and its family members’ physiological functions and Swedish mutant APP (APP_swe_)’s pathological roles in muscles and NMJ. Understanding APP’s pathophysiological functions in muscles and NMJ is likely to uncover insights not only into neuromuscular diseases but also AD. We summarize key findings from the burgeoning literature, which may open new avenues to investigate the link between muscle cells and brain cells in the development and progression of AD and sarcopenia.

## 1. Introduction

Sarcopenia is a condition characterized by progressive muscle atrophy with a loss of muscle strength and/or function during age, which was first defined by the European Working Group on Sarcopenia in 2010 [1]. The incidence of sarcopenia has attracted increasing attention, as it raises the risk of falls and disability and impairs the quality of life in the old. The knowledge of sarcopenia is also constantly renewed from a muscle-centered perspective to muscle-to-neuron/brain crosstalk. It not only emphasizes the role of neuronal excitability and NMJ signaling in regulating muscular contraction but also highlights the impact of retrograde signaling from muscles to motor neurons. Interestingly, sarcopenia is often associated with patients with neurodegenerative diseases, including AD, ALS, and PD [2,3,4,5], raising a question of how muscle-relevant sarcopenia is linked with brain/neuron degenerative diseases. While epidemic studies have demonstrated a strong association of sarcopenia with neurodegenerative diseases, it remains unclear how they are associated and what their relationship(s) are. APP is an integral membrane protein that is concentrated at the synapse, not only in the CNS but also in the NMJ, implicating APP as a shared molecule that underlies the development of neurodegeneration diseases and sarcopenia. Below, we summarized APP’s physiological functions in muscle and the NMJ and discussed mutant APP’s potential contributions to the development of sarcopenia and neurodegenerative diseases.

## 2. AD Association with Sarcopenia

AD, the most common type of dementia, affects about ten percent of the population over 65 years of age [6]. It is estimated that there are about 6.7 million people in the USA who have AD, and this number could grow to 13.8 million by 2060 [7]. Patients with AD suffer from declined memory, cognitive deficits, and changes in personality. In the past, AD was diagnosed for certain only after death when histopathology revealed plaques and tangles in the brain. Now, people can diagnose Alzheimer’s disease by biomarkers that can also detect the presence of plaques and tangles in cerebrospinal fluid (CSF) or blood. People find that the development of AD always obscurely progresses for many years, even decades, before being noticed. The time from obscure onset to clinical diagnosis is called the preclinical phase, a time in which the recent literature has noted there seems to be a systemic or multi-organ disorder that affects not only the brain but also peripheral tissues in AD [8].

Sarcopenia is defined as a progressive skeletal muscle disorder with the accelerated loss of muscle mass and function. Sarcopenia is also a multifactorial disorder, similar to AD. It commonly occurs in older people during aging, influenced by genetic and lifestyle factors [9,10]. It is characterized by a combination of measurements of muscle mass, muscle strength, and physical performance. Usually, sarcopenia is suspected when low muscle strength is detected. The diagnosis of sarcopenia can be confirmed by the additional presence of low muscle quantity or quality. If low physical performance is also associated with sarcopenia patients, it will be considered severe sarcopenia [10]. The changes in muscles (e.g., muscle fiber size and number) are obviously detectable at the microscopic level. Muscle fiber-type transformation is also visible in sarcopenia patients [11]. Although the exact causes of sarcopenia are not clear, it is believed that multiple factors, including aging, chronic inflammation, malnutrition, such as low protein/energy intake, malabsorption, and inactivity, such as bed rest, immobility, and sedentary lifestyle, are involved in the development of sarcopenia [9].

Interestingly, AD patients, even at early stages, showed a high prevalence rate of sarcopenia [12], leading to severe body mass loss and, specifically, a decrease in muscle strength. Recent multiple systemic and meta-data analyses also demonstrate the association between sarcopenia and cognitive impairment independently of sex, population demographics, and the definition of sarcopenia [13,14]. Decreased skeletal muscle mass in older adults with AD is related to AD progression [15]. Conversely, increased muscle mass is associated with a lower probability of AD in elderly subjects [16].

Moreover, emerging evidence indicates that AD is associated with changes in muscle mass but also in muscle strength. In addition, reduced muscle strength or frailty is also related to the risk of developing AD. On the other hand, larger muscle strength is related to a smaller risk of mild cognitive impairment (MCI), a believed precursor to AD, and is inversely related to brain atrophy or hippocampal volume in patients with AD [17,18], larger handgrip strength associated with larger hippocampal volumes. It is of interest to notice that a recent prospective cohort study, with 340,212 participants, suggests an association of an absolute 5 kg increment of grip strength with a lower risk of AD (HR 0.874) [19]. Another prospective cohort study, with 495,700 participants with a median 12 years follow-up, demonstrates that walking pace is inversely associated with new-onset dementia, particularly in young participants with lower APOEε4 dosage [20]. However, the mechanisms underlying the association between cognitive impairment and sarcopenia appear to be complicated. Loss of cholinergic neurotransmission significantly contributed to the deterioration in cognitive function in AD. Acetylcholine (Ach) is a major neurotransmitter at the NMJ, and muscle atrophy has been reported when responding to cholinergic deficits [21]. In addition, both AD and sarcopenia share similar environmental risk factors or pathological processes, such as aging, hormone dysfunction, chronic inflammation, oxidative stress, and mitochondrial dysfunctions [22]. Furthermore, the frequent causes of sarcopenia, including physical inactivity or malnutrition, are also commonly occurring in AD patients, although some studies found that the physical activity and nutrient levels measured by serum albumin and serum total protein were not associated with sarcopenia in AD [23]. Moreover, muscles are believed to function as an endocrine organ, expressing and secreting numerous myokines, including cytokines (e.g., IL6), peptides (e.g., FNDC5/Irisin) [24,25], and growth factors (e.g., BDNF [26] and IGF1 [27,28]). Many of these myokines can be regulated by exercise and play important roles in regulating cognitive function. Furthermore, AD and sarcopenia share some genetic risk factors, such as APOE [29] and APP [30]. Recently, some groups have identified susceptibility loci of muscle weakness in older men and women in a genome-wide association study meta-analysis [31]. It included 256,523 people from 22 cohorts. Interestingly, they found that Alzheimer’s disease had a positive genetic correlation with low muscle strength. Further, they also discovered the increased likelihood of muscle weakness with genetically predicted depression, which is a common syndrome in Alzheimer’s disease, under Mendelian randomization analysis [31]. 

It is known that muscle contraction requires proper functions of the NMJ. Muscle weakness in AD may result from reduced muscle mass and/or deficits at the NMJ. Interestingly, AD patients could have decreased muscle strength without loss of muscle mass [32], suggesting that the reduced muscle strength in AD is independent of the decreased muscle mass. However, the NMJ function in AD patients remains little investigated. Therefore, a scrupulous investigation of the causal relationship between NMJ and AD is far from being established and is an important issue to be addressed in future research. 

## 3. APP Family Proteins in Skeletal Muscles and NMJ

### 3.1. APP Family Proteins and Their Expressions in Skeletal Muscles and NMJ

APP is a single transmembrane protein containing a large extracellular glycosylated N-terminus and a shorter C-terminus intracellular domain (Figure 1). APP and two other APP-like proteins (APLP1 and APLP2) belonging to the APP family have a similar structure and membrane topology as APP. While they are abundantly expressed in the brain/neurons, they are detectable in various organs or peripheral tissues, including skeletal muscles and the NMJ [33]. In the NMJ, APP and APLP2 are highly expressed in both presynaptic and postsynaptic membranes; APLP1 is only expressed in the presynaptic membranes [34]. In muscles, APP can be detected in the cytoplasm at embryonic day 16 (E16). Until at birth (P0), APP begins to be localized at the NMJ. By postnatal day 5 (P5), APP expression increases and co-localizes with other NMJ proteins, such as the acetylcholine receptors, as concentrated in the NMJ [35].

### 3.2. APP Cleavage Products

APP is a well-studied transmembrane protein that undergoes non-amyloidogenic or amyloidogenic processing. In the pathway of non-amyloidogenic processing, the APP protein is cleaved first by α-secretase (ADAMs) and then γ-secretase, such as Presenilin1/2, Presenilin enhancer 2 (PEN2), etc. An N-terminal sAPP-α fragment, a P3 peptide, and the APP intracellular domain (AICD) are thus generated (Figure 1). Interestingly, the sAPP-α fragment is believed to be neuroprotective in the NMJ [36]. In the pathway of amyloidogenic processing, APP is cleaved first by the β-secretase and then γ-secretase to release the sAPP-β fragment, the Aβ1–40/42 peptides, and an AICD (Figure 1). The amyloid beta (Aβ) can only be generated from APP, but not its other family members, which is believed to be an important culprit in AD development, and toxic not only to neurons but also skeletal muscle [37,38,39].

### 3.3. APP Family Proteins’ Functions in NMJ

APP is required for NMJ formation. Although APP knockout mice are viable, they exhibit growth deficits with reduced body weight and decreases in grip strength and circadian locomotor activity [40,41]. These phenotypes imply there may exist a defect of neuromuscular transmission or the loss of muscle contraction [41]. Indeed, double knockout (dKO) mice of APLP2/APP die at P0 with abnormal NMJ formation [42,43] (Figure 2A,B). There are significant presynaptic deficits, including excessive nerve terminal sprouting and reduced synaptic vesicles with functional defects of synaptic transmission (a defective frequency of mEPP while preserving amplitude) [44]. Interestingly, the expression of the sAPP-α domain can rescue the postnatal death of the APLP2/APP dKO mice (which still have both pre and postsynaptic deficits) [36] and can also rescue motor dysfunction and the reduced body weight in APP single-knockout mice [41]. These observations suggest that APP/APLP2 plays an essential role in NMJ function and formation, and different domains of APP seem to have different functions in the NMJ.

Notice that APP is expressed and distributed at both motor neurons and muscles with a 1:1 stoichiometry at the presynaptic and postsynaptic sites [44], implicating potential trans-synaptic APP interactions as a functional modality. In line with this view are observations that both neuronal-dKO (neuronal APP cKO in the APLP2 global KO background) and muscle-dKO (muscle APP cKO in the APLP2 global KO background) mice exhibit neuromuscular phenotypes as those of germline APP/APLP2 dKO mice [44]. Moreover, compared with the neuronal-dKO, the muscle-dKO mice show a more severe NMJ phenotype, such as decreases in presynaptic choline transporter (CHT) targeting, that control the transmission rate of the cholinergic synapse in both the NMJ and CNS. Postsynaptic APP deletion also led to a deficit in spontaneous vesicle release [44], suggesting a critical role of postsynaptic APP in regulating presynaptic function. In agreement, our lab has created a transgenic mouse with muscle-specific overexpression of APP_swe_, which showed similar presynaptic NMJ phenotypes as those of muscle-dKO mice (unpublished results) (Figure 2C), suggesting a dominant negative effect of the muscle APP_swe_. Further investigations to test this view are necessary.

### 3.4. APP’s Binding Partners in NMJ

As an integral membrane protein, APP is implicated as a ligand and/or receptor for NMJ formation. Recent studies have revealed its potential binding partners at the NMJ. The APP ectodomain can interact with the ligand binding domain of low-density lipoprotein receptor-related protein 4 (LRP4), thereby inducing the phosphorylation of Musk for AChR clustering on its own or cooperatively with agrin [45] (Figure 3). The LRP4 LDLa domain and APP E1 domain are critical sites for their interaction [46]. The ectodomain of APP can also interact with agrin, although which domain that binds to agrin is not well characterized. This interaction has been shown to regulate APP function to underlie its role in the development of the NMJ in mice. APP can also interact with the choline transporter (CHT) through APP’s C-terminal domain (Figure 3), which is critical for CHT endocytic recycling [47]. The loss of APP results in defective CHT localization and also decreased CHT activity, contributing to the NMJ deficit [47]. Additionally, DR6 is also named the TNF receptor superfamily member 21 (TNFRSF21)). It is a membrane receptor with a Death domain in the cytoplasm. The amino terminus of APP, released from sAPPβ after being further cleaved, is reported to be a DR6 (death receptor 6) ligand. The amino terminus of APP, as an active ligand, activates DR6, inducing degeneration of cell bodies and axons degeneration via different caspase enzymes (caspase-3 and caspase-6, respectively [44,48]. The DR6 KO mice show similar NMJ phenotypes with that of APP/APLP2 DKO mice, supporting the role of the DR6 interaction with APP/APLP2 in vivo [44,48]. It was reported that APP could be transported in vesicles [49]. These vesicles also transported some synaptic proteins such as synapsin-1 and contained active α secretase (ADAM10), which allowed the release of APP at the synaptic terminal [50,51]. Co-transported synapsin-1 and APP may relate to the balance of the reserve/mature vesicles. APP is also reported to act as a novel synaptic adhesion protein, recruiting Mint1 and Cask [44] or Mint2 and Munc18 [36] to form a presynaptic complex for NMJ function. Munc18 is an important protein for exocytosis at active zones [52], and therefore APP may also take part in different pools of synaptic vesicle transition. Moreover, APP is shown to bind to extracellular matrix proteins, such as laminin, heparin, collagen, Ca^2+^ channels, etc., which may also be involved in APP’s function in the NMJ. In aggregate, there are many APP-binding proteins in the NMJ; further investigating their functional significance may uncover important insights into APP’s role in NMJ development, maintenance, and aging.

## 4. APP and Aβ in AD and Sarcopenia Development

### 4.1. APP and Aβ in AD Development

Many genetic risk genes/loci have been identified in patients with early-onset (EO) and late-onset (LO) AD. Among them, *App*, a Mendelian gene for the EOAD, has been studied extensively. Aβ, a major marker and culprit of AD, is created by β-and γ-secretases/proteases under the proteolytic processes of APP [53]. The APP mutations identified in EOAD patients (e.g., Swedish mutations, APP_swe_) increase the level of Aβ by favoring proteolytic cleavage by β- and γ-secretases [54]. Similar effects are observed for EOAD mutations of presenilin 1/2 (PS1/2), a major component of the γ-secretase [55]. These studies provide genetic support for the amyloid Aβ hypothesis. Notably, the Aβ levels in the brain are elevated not only in EOAD but also in LOAD patients without APP/PS1/2 mutations [56]. Other genetic risk factors for LOAD, including the ε4 allele of ApoE, TREM2-mutation, CD33-upregulation, SorL1-, and VPS35-deficiency, also increase the Aβ levels by promoting its production and/or impairing Aβ clearance [57,58,59]. Besides the amyloidogenic processing pathway of APP that contributes to AD, the APP family also plays a role in neuronal function, including neuronal excitability and synaptic plasticity, which is important for memory in adulthood [60]. APP can be found in both the somatodendritic and the axonal compartments of neurons. In axons, they are mainly transported in vesicles and enriched at active zones [61]. The APP family is both responsible for the defect in memory and synaptic plasticity at the pre and postsynaptic levels [60]. It was found with the increase in paired-pulse facilitation (PPF) and synaptic facilitation and also reduced postsynaptic NMDA receptor-mediated responses [60] in a conditional triple KO (cTKO) transgenic mouse. These results were also in line with the phenotype of APP/ APLP2 dKO mice [62]. The decrease in Kv7 channels, meaning the decrease in the M-type potassium current, might be the underlying molecular mechanism for the defect in memory and LTP in these conditional triple KO (cTKO) transgenic mice [60]. All these data provided new insights into the function of the APP family in the CNS during adulthood, which may also contribute to the development of AD. Further studies will be necessary to relieve the possible molecular mechanism.

### 4.2. APP and Aβ in the Development of Sarcopenia-like Phenotype

Interestingly, muscle strength and mass are reduced in APP_swe_/Aβ-based AD animal models, such as Tg2576, a transgenic AD mouse model that expresses human APP with Swedish mutations (K670N/M671L) under the control of the hamster prion promoter, and thus expressing APP_swe_ ubiquitously. Remarkably, muscle weakness and lower muscle mass occur at the young adult age of Tg2576, months before any brain defects can be detected. In addition to the reduced muscle mass and strength, increased NMJ denervation and fragments are detected with aging [63,64]. However, the underlying mechanism remains less well understood.

### 4.3. Muscular Swedish Mutant APP/Aβ’s Contributions to Not Only Sarcopenia-like Deficit but Also AD-Relevant Brain Pathology

Interestingly, there was also evidence for cognitive deficits in a few myopathies. Although myopathies represent a wide spectrum of heterogeneous muscular diseases, which can be hereditary or acquired, there were different causal classifications of myopathy found with cognitive deficits, even white matter lesions in the brain [65], such as Duchenne muscular dystrophy (DMD) [66]. Duchenne muscular dystrophy (DMD) is a hereditary muscle disorder caused by the mutation of the dystrophin gene. Some studies found boys with DMD were more sensitive to learning disabilities; their verbal IQ was more affected, and up to 40% DMDs had deficits in short-term verbal memory [67], whilst the mdx mice also displayed impairment in short-term memory [68]. There was also exited Alzheimer’s disease-like neuropathology: an increased level of soluble amyloid precursor protein β (APPβ) and APP and a decreased level of soluble APPα, suggestive of a shift towards amyloidogenesis in APP processing, which is similar to Alzheimer’s disease [69]. However, there is also a similar change in APP processing in muscles not mentioned. This indicates that the muscle–brain axis takes part in the development of AD. Furthermore, APP is the initially identified risk gene for early-onset (EO) AD, which contains more than 50 pathogenic mutations, including APP duplication and APP missense mutation [70]. Among the APP mutations, the Swedish mutant, which is a double mutation at codons 670 and 671, identified in a Swedish family in 1992, has been well-studied [71]. It is believed that these mutations in APP contribute to EOAD by increasing the Aβ levels in the brain, as these mutations located at the N-terminus of the β-secretase cleavage site promote abnormal cleavage and Ab production [72]. However, whether and how APP_swe_ in muscle contributes to the development of Sarcopenia or AD remains less well-studied and is little summarized.

Notice that in AD patients’ muscles, the levels of Aβ_40_ and Aβ_42_ are both evaluated, as compared to the control muscle samples. While the muscular Aβ_40_ levels are correlated with age, the muscular Aβ_42_ levels are associated with the Braak stages of AD disease [73]. Additionally, there are reports about mitochondrial abnormalities [74,75,76] that lead to impaired Ca^2+^ release and contractility [39] in muscles with APP overexpression or APP mutant. These observations suggest that the dysfunctional APP or Aβ overproduction is a potential common risk factor for muscle and brain degeneration. However, their relationships and whether APP_swe_-induced muscle deficits contribute to AD-relevant brain pathology remain largely unknown.

Using TgAPP_swe_^HSA^, a mouse model that expresses APP_swe_ specifically in muscles, we detected not only sarcopenia-like muscle deficits but also age-dependent behavior abnormality and brain pathology, such as glial cell activation, brain inflammation, and blood–brain barrier disruption, which are largely in the hippocampus [30] (Figure 4). We further show that the APP_swe_ expression in muscles induces senescence and expressions of senescence-associated secretory phenotypes (SASPs), which cause chronic systemic inflammation [30] (Figure 4). Inhibition of the senescence diminishes nearly all the phenotypes, including behavior changes and brain pathology in TgAPP_swe_^HSA^ mice [30]. These results demonstrate the role of muscular APP_swe_ in AD or sarcopenia development and uncover the crosstalk from muscles to the brain [30]. Moreover, of interest to note are the previous reports that target muscle therapy by myostatin knockdown in muscles that can ameliorate not only the motor function but also cognitive function in AD mice [77]. These observations raise interest in a potential therapeutic strategy development for AD in future investigations.

## 5. APP/Aβ’s Contribution to Amyotrophic Lateral Sclerosis (ALS)

In addition to AD, APP/Aβ is reported to be involved in the development of ALS. ALS is a fatal degenerative disease of the motor neurons with rapid progression. ALS is clinically heterogeneous in terms of syndromes, genetics, prognosis, different motor neuron burden pattern, and pathophysiology. However, the muscle MRI appears to be a practical biomarker to evaluate the development of motor neuron diseases objectively under systematic review [78]. The muscle MRI measures positively correlate with disease severity [3,78]. This indicates the association between ALS and sarcopenia. Recent studies indicated the “dying-back” phenomenon in the early stages of ALS, even in different ALS-related mutant mice, prior to motor neuron death. Therefore, NMJ dysfunction contributes not only to the pathogenesis of ALS but also to the pathology of sarcopenia. The underlying molecular mechanism remains little known. Although APP is not the risk gene for ALS, there are reports of abnormal APP processing in patients with ALS [79,80]. SOD1 mutation, a well-known risk factor for ALS, induces APP conformational change and increases the APP levels in both spinal cords and muscles of ALS [80,81,82]. Interestingly, the significant increase of APP is initially detected in the muscles but not the spinal cord, preceding the onset of ALS disease [81]. Genetic ablation of APP can significantly delay motor function decline and improve innervation of the NMJ, muscle contraction, and motor neuron survival [83]. In line with this, the APLP2 expression was increased in the spinal cord in the ALS model. APLP2 knockout in ALS mice can significantly delay disease progression and increase the time of survival. This suggests that the interaction of muscular APP and APP/APLP2 in motor neurons may contribute to the survival of motor neurons in ALS [84]. Aside from this, inhibition of the APP beta-secretase cleavage can also delay ALS progression in mice [85]. These observations suggest that the modulation of APP/Aβ levels might be beneficial in ALS therapy. However, how muscular APP contributes to ALS development needs to be further investigated.

## 6. APP/Aβ’s Contribution to Myopathy

As we have mentioned before, myopathies represent heterogeneously, but several different causal classifications of myopathies were found with disease hallmarks, such as an aggregation of Aβ, including sporadic inclusion body myositis (sIBM), GNE myopathy (GNEM) and chloroquine-induced myopathy. This indicated the important role of APP in the physiology and pathology function of muscle cells.

### 6.1. APP/Aβ’s Contribution to Sporadic Inclusion Body Myositis (sIBM)

Notably, sIBM is the most common inflammatory myopathy in the elderly. It is characterized by progressive weakness of muscles, especially those of the arms and legs, leading to grip strength deficits and gait disturbance. Interestingly, Aβ protein deposition is one of the classical sIBM pathologic features. A proteomic study of sIBM in patients’ muscle samples also showed increased APP expression [86]. The overexpression of beta APP in skeletal muscles induces an age-dependent development of histopathological characteristics of sIBM, such as intracellular betaAPP aggregation, muscle-centric nuclei, and inflammation [87,88], suggesting a role of APP in sIBM development. Although some studies have indicated the involvement of autophagy in APP/Aβ in sIBM [89], little is known about the trigger of protein deposition and inflammation. Further studies in APP metabolic processing in physiology and pathology may help to uncover this mystery.

### 6.2. APP/Aβ’s Contribution to GNE Myopathy (GNEM)

GNEM, a neuromuscular disorder caused by mutations in GNE (UDP N-acetylglucosamine 2 epimerase/N-acetylmannosamine kinase), leads to muscle degeneration with age. In GNEM, the main pathological feature is the formation of a rimmed vacuole in muscle, which includes aggregated proteins such as Aβ and tau [90]. Degradative products from membranes and mitochondria in inclusion bodies can also be found under electron microscopy [91]. There was some evidence showing that mRNA expression of APP was not statistically significant between GNEM and the control [92], suggestive of APP metabolic processing playing a part in it. GNE is a major enzyme for sialic acid biosynthesis, where the mutation in GNE affects the sialylation of glycoproteins. Several proteins that play a role in Aβ deposition can undergo glycosylation, including APP [93]. N-glycosylation of APP is related to protein secretion, trafficking, and non-amyloidogenic processing [94,95]. However, the glycosylation patterns of APP or Aβ production were not studied in GNEM. Thus, there is a need to investigate the physiology or pathology function of APP in muscle cells.

### 6.3. APP/Aβ’s Contribution to Chloroquine-Induced Myopathy

Chloroquine, a potent lysosomotropic agent, induces myopathy in experimental animals, such as rimmed vacuole (RV) myopathy in humans. The abnormal accumulation of amyloid beta protein (Aβ), mainly Aβ42, had been demonstrated in denervated muscle fibers in chloroquine-induced myopathy in rats [96,97]. However, how chloroquine-induced injury leads to the change in APP metabolic processing remains largely unknown.

## 7. Conclusions

APP, a well-investigated protein for AD development, is widely expressed across many tissues, including skeletal muscle. APP metabolism and its function in muscles are crucial for synapse development and maintenance at the NMJ. APP/Aβ pathology in muscles is strongly associated with myopathy in neurodegenerative disorders, including AD and ALS. However, the molecular mechanisms that link muscular APP to the brain’s pathology and function remain elusive. Identification of the role of APP in muscles will also establish the link between sarcopenia and neurodegenerative diseases. These may lead to insights into muscle–brain crosstalk and suggest potential contributions to age-related degeneration.

## Figures and Tables

**Figure 1 ijms-24-07809-f001:**
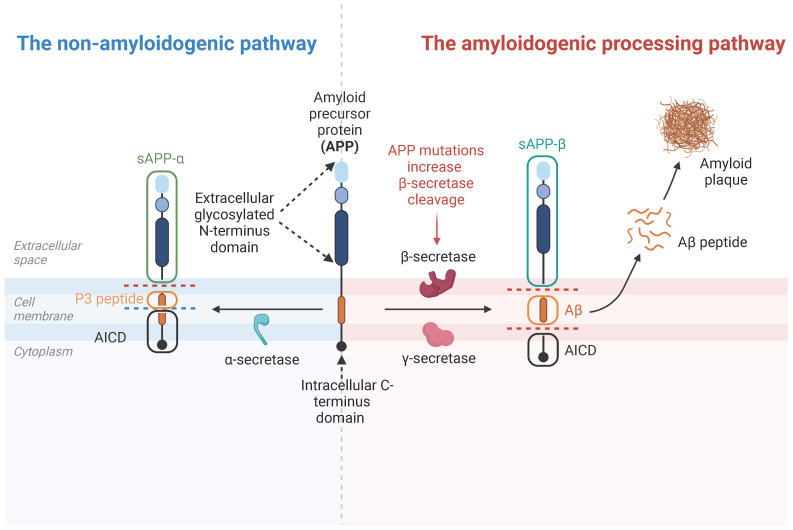
APP structure and its cleavage products. Abbreviation: AICD, APP intracellular domain. (Created with BioRender.com, accessed on 24 March 2023).

**Figure 2 ijms-24-07809-f002:**
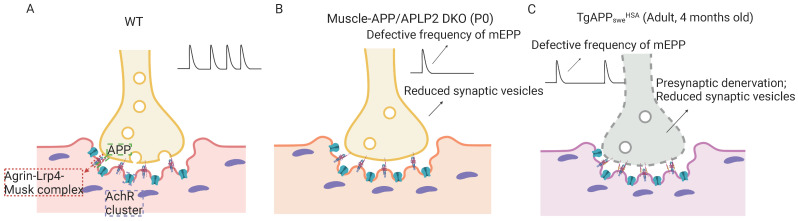
Schematic illustration of the NMJ in WT (**A**), muscle-specific APP/APLP2 double knockout mice (M-dKO) (**B**), and muscle-specific human APP_swe_ expression mice (TgAPP_swe_^HSA^) (**C**), The NMJ phenotypes in the M-dKO and TgAPP_swe_^HSA^ are indicated. Abbreviations: AChR, acetylcholine receptor; APP, amyloid precursor protein; mEPP: miniature end-plate potential; NMJ, neuromuscular junction. (Created with BioRender.com, accessed on 24 March 2023).

**Figure 3 ijms-24-07809-f003:**
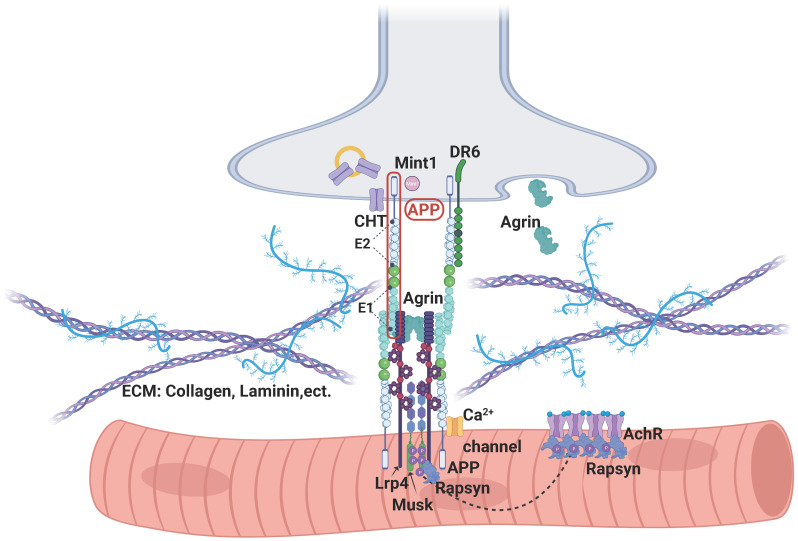
Molecular interaction of APP/APLPs at the NMJ. Shown is the NMJ in a single muscle fiber. APP/APLPs are expressed in both presynaptic and postsynaptic membranes. The extracellular domain of APP includes E1 and E2 domains. E1 domain interacts with the LDLa domain of LRP4, thereby inducing the phosphorylation of Musk for AChR clustering by itself or cooperatively with agrin. APP extracellular domain can also interact with ECM proteins, such as collagen and laminin. The intracellular domain of APP can interact with CHT, DR6, Ca^2+^ channels, etc. The complex and diverse binding partners of APP in NMJ may contribute to its functions in NMJ. Abbreviations: AChR, acetylcholine receptor; APP, amyloid precursor protein; CHT, choline transporter; DR6, death receptor 6; ECM, extracellular matrix; Lrp4, low-density lipoprotein receptor-related protein; MuSK, muscle-specific kinase; NMJ, neuromuscular junction. (Created with BioRender.com, accessed on 24 March 2023).

**Figure 4 ijms-24-07809-f004:**
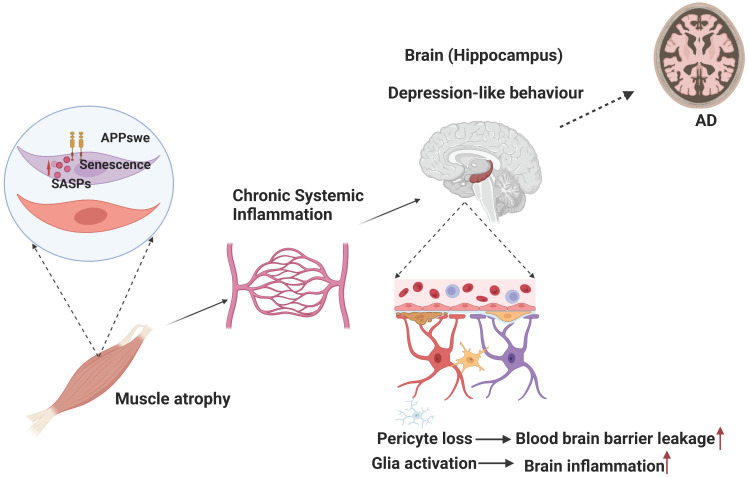
Muscular APP_swe_ may contribute to AD development. APP_swe_ overexpression in muscles induces senescence and expressions of senescence-associated secretory phenotypes (SASPs), which induce chronic systemic inflammation, leading to behavior abnormalities and brain pathologies, such as blood-brain barrier disruption and neuroinflammation caused respectively by pericyte loss and glial cell activation. Abbreviations: APP_swe_, APP Swedish mutant. (Created with BioRender.com, accessed on 24 March 2023).

## Data Availability

Not applicable.

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
