# Peer review of "APP in the Neuromuscular Junction for the Development of Sarcopenia and Alzheimer’s Disease"

_ijms, 2023, doi:10.3390/ijms24097809_

Round 1
Reviewer 1 Report
This is an excellent review that highlights comprehensive knowledge of the literature in the field. The manuscript is well-written, and the most important references that highlight ideas in the field are cited.
Author Response
We sincerely thank the reviewer for their positive evaluation.
Reviewer 2 Report
The review of Wu et al. summarizes existing knowledge on the role of APP in NMJ and its possible relation to sarcopenia. Although AD is related to sarcopenia, the molecular background of this relation is largely unknown. Current insights are mainly based on the Swedish mutant APP, published recently. The manuscript of Wu et al. gives, however, a broader insight on the APP interacting partners in NMJ and provides relation of APP to other neuromuscular diseases.
The introduction gives an overview on the topic, although the existing multifactorial nature of sarcopenia should be highlighted. Illustrations are of sufficient quality and facilitate the understanding of text. The text is logical and clear, illustrating different aspects of the topic.
Comments:
Line 46-47 please provide a source for the statistics.
Sarcopenia is multifactorial. Please mention also other causes of sarcopenia, apart from the molecular, such as malnutrition or physical inactivity.
Please discuss shortly synaptic localisation of APP in the brain
Line 254-only in a few myopathies there is a brain involvement.
Line 257 „even lesions in brain” – please use the MRI characteristics, otherwise it is not precise enough.
I found the following manuscript a valuable source of information of APP role in IBM https://doi.org/10.1186/s12953-014-0045-2
For the clarity, I would avoid starting sentences with “But” and “And”
Author Response
A point-by-point reply to the reviewer can be found below.
- Line 46-47 please provide a source for the statistics.
Yes, provided (see Line 47-48 in revised review).
- Sarcopenia is multifactorial. Please mention also other causes of sarcopenia, apart from the molecular, such as malnutrition or physical inactivity.
Agree! These are good points, which have been included in revised review (see Line 58-71).
- Please discuss shortly synaptic localisation of APP in the brain.
Thanks for this suggestion. This point has been added in line 253-254.
- Line 254-only in a few myopathies there is a brain involvement.
Revised as suggested (see Line 279).
- Line 257 „even lesions in brain” – please use the MRI characteristics, otherwise it is not precise enough.
It has been revised as suggested (see Line 282).
- I found the following manuscript a valuable source of information of APP role in IBM https://doi.org/10.1186/s12953-014-0045-2
Thanks for the information. We have added this point and reference (see Line 370-371).
- For the clarity, I would avoid starting sentences with “But” and “And”
Again, thanks for the suggestions. We have revised these descriptions in the revised manuscript.

Reviewer 3 Report
The manuscript is interesting, comprehensive and has merit. I only have a few minor comments.
1. The authors should inlcude a definition and diagnostic criteria of sarcopenia.
2. The authors could inlcude a short description how sarcopenia and muscle atrophy affect skeletal muscle fibres on microscopic level.
3. Is there any estimate how potent is AD as an indpendent risk factor for sarcopenia, standardised for loss of physical activity and weight loss.
Author Response
A point-by-point reply to the reviewer can be found below.
- The authors should inlcude a definition and diagnostic criteria of sarcopenia.
We thank the reviewer for pointing out this important issue. We have included this point and diagnostic criteria of sarcopenia (see Line 59-72).
- The authors could inlcude a short description how sarcopenia and muscle atrophy affect skeletal muscle fibres on microscopic level.
Yes, We have added this point in line 67-69.
- Is there any estimate how potent is AD as an indpendent risk factor for sarcopenia, standardised for loss of physical activity and weight loss.
This is a very important question, which needs more efforts to investigate. As we have mentioned, both AD and sarcopenia share similar environmental risk factors or pathological processes. The frequent causes of sarcopenia including physical inactivity or malnutrition are also commonly occurred in AD patients, which would further burden the risk of sarcopenia in AD. However, some studies report that the physical activity and nutrient levels measured by serum albumin and serum total protein were not associated with sarcopenia in AD[1]. Some studies also found that muscle strength or physical performance, the two major diagnostic criteria of sarcopenia, are inversely associated with new-onset dementia[2,3]. Other studies, genome-wide association, have uncovered the link of AD with sarcopenia [4]. We have revised the manuscript to include these points in Line 98-102.
Reference:
- Kimura, A.; Sugimoto, T.; Niida, S.; Toba, K.; Sakurai, T. Association Between Appetite and Sarcopenia in Patients With Mild Cognitive Impairment and Early-Stage Alzheimer's Disease: A Case-Control Study. Frontiers in nutrition 2018, 5, 128, doi:10.3389/fnut.2018.00128.
- Kuo, K.; Zhang, Y.R.; Chen, S.D.; He, X.Y.; Huang, S.Y.; Wu, B.S.; Deng, Y.T.; Yang, L.; Ou, Y.N.; Guo, Y.; et al. Associations of grip strength, walking pace, and the risk of incident dementia: A prospective cohort study of 340212 participants. Alzheimer's & dementia : the journal of the Alzheimer's Association 2022, doi:10.1002/alz.12793.
- He, P.; Zhou, C.; Ye, Z.; Liu, M.; Zhang, Y.; Wu, Q.; Zhang, Y.; Yang, S.; Xiaoqin, G.; Qin, X. Walking pace, handgrip strength, age, APOE genotypes, and new-onset dementia: the UK Biobank prospective cohort study. Alzheimer's research & therapy 2023, 15, 9, doi:10.1186/s13195-022-01158-6.
- Jones, G.; Trajanoska, K.; Santanasto, A.J.; Stringa, N.; Kuo, C.L.; Atkins, J.L.; Lewis, J.R.; Duong, T.; Hong, S.; Biggs, M.L.; et al. Genome-wide meta-analysis of muscle weakness identifies 15 susceptibility loci in older men and women. Nature communications 2021, 12, 654, doi:10.1038/s41467-021-20918-w.
